# Cortical encoding of melodic expectations in human temporal cortex

Giovanni M Di Liberto[1]*, Claire Pelofi[2,3†], Roberta Bianco[4†], Prachi Patel[5,6], Ashesh D Mehta[7,8], Jose L Herrero[7,8], Alain de Cheveigné[1,4], Shihab Shamma[1,9]*, Nima Mesgarani[5,6]*

[1]Laboratoire des systèmes perceptifs, Département d'études cognitives, École normale supérieure, PSL University, CNRS, 75005 Paris, France; [2]Department of Psychology, New York University, New York, United States; [3]Institut de Neurosciences des Système, UMR S 1106, INSERM, Aix Marseille Université, Marseille, France; [4]UCL Ear Institute, London, United Kingdom; [5]Department of Electrical Engineering, Columbia University, New York, United States; [6]Mortimer B Zuckerman Mind Brain Behavior Institute, Columbia University, New York, United States; [7]Department of Neurosurgery, Zucker School of Medicine at Hofstra/Northwell, Manhasset, United States; [8]Feinstein Institute of Medical Research, Northwell Health, Manhasset, United States; [9]Institute for Systems Research, Electrical and Computer Engineering, University of Maryland, College Park, United States

**Abstract** Humans engagement in music rests on underlying elements such as the listeners' cultural background and interest in music. These factors modulate how listeners anticipate musical events, a process inducing instantaneous neural responses as the music confronts these expectations. Measuring such neural correlates would represent a direct window into high-level brain processing. Here we recorded cortical signals as participants listened to Bach melodies. We assessed the relative contributions of acoustic *versus* melodic components of the music to the neural signal. Melodic features included information on pitch progressions and their tempo, which were extracted from a predictive model of musical structure based on Markov chains. We related the music to brain activity with temporal response functions demonstrating, for the first time, distinct cortical encoding of pitch and note-onset expectations during naturalistic music listening. This encoding was most pronounced at response latencies up to 350 ms, and in both *planum temporale* and *Heschl's gyrus*.

*For correspondence:
diliberg@tcd.ie (GMDL);
sas@isr.umd.edu (SS);
nima@ee.columbia.edu (NM)

†These authors contributed equally to this work

Competing interests: The authors declare that no competing interests exist.

## Introduction

Experiencing music as a listener, performer, or a composer is an active process that engages perceptual and cognitive faculties, endowing the experience with memories and emotion (*Koelsch, 2014*). Through this active auditory engagement, humans analyze and comprehend complex musical scenes by invoking the cultural norms of music, segregating sound mixtures, and marshaling expectations and anticipation (*Huron, 2006*). However, this process rests on the 'structural knowledge' that listeners acquire and encode through frequent exposure to music in their daily lives. Ultimately, this knowledge is thought to shape listeners' expectations and to determine what constitutes a 'familiar' musical style that they are likely to understand and appreciate (*Morrison et al., 2008*; *Hannon et al., 2012*; *Pearce, 2018*). There is convincing evidence that musical structures can be learnt through the passive exposure of music in everyday life (*Bigand and Poulin-Charronnat, 2006*; *Rohrmeier et al., 2011*), a phenomenon that was included in several models musical learning. The

connectionist model from Tillman and colleagues, which simulated implicit learning of pitch structures occurring in Western music, was shown to successfully predict behavioral and neurophysiological results on music perception (*Tillmann et al., 2000*). Further developments led to models that accurately reflect listeners' expectations on each upcoming note by considering both the listener's musical background (long-term model) and proximal knowledge (short-term model) (*Pearce, 2005*; *Huron, 2006*).

A similar learning process has been demonstrated in many domains of human learning (e.g., language; *Saffran et al., 1997*; *Toro et al., 2005*; *Finn et al., 2014*), as well as in animals that must learn their species-specific songs and vocalizations (*Woolley, 2012*). Theoretically, this is often conceptualized as learning the statistical regularities of the sensory environment (*Romberg and Saffran, 2010*; *Erickson and Thiessen, 2015*; *Bretan et al., 2017*; *Skerritt-Davis and Elhilali, 2018*) so as to predict upcoming events, in a process referred to as 'statistical learning'. Supporting evidence is that failed predictions due to deviations in the sensory sequence produce measurable brain activations that have been associated with the detection and strength of these irregularities, that is prediction error (*Vuust et al., 2012*; *Clark, 2013*; *Moldwin et al., 2017*; *Omigie et al., 2019*; *Quiroga-Martinez et al., 2019b*; *Quiroga-Martinez et al., 2019a*), and with learning (*Storkel and Rogers, 2000*; *Attaheri et al., 2015*; *Qi et al., 2017*).

Decades of research have established that listeners have strong and well-defined musical expectations (*Schmuckler, 1989*; *Cuddy and Lunney, 1995*; *Margulis, 2005*; *Morgan et al., 2019*) which depend on a person's musical culture (*Carlsen, 1981*; *Kessler et al., 1984*; *Krumhansl et al., 2000*; *Eerola et al., 2009*). Neurophysiology studies demonstrated that violation of music expectations elicits consistent event-related potentials (ERPs), such as the mismatch negativity (MMN) or the early right-anterior negativity (ERAN), which typically emerges between 100 and 250 ms after the onset of the violation (*Koelsch, 2009*; *Vuust et al., 2012*). Such brain responses have been measured for a range of violations of auditory regularities, including out-of-key notes embedded in chords (e.g. Neapolitan sixth in *Koelsch et al., 2000*), unlikely chords (e.g. double dominant in *Koelsch et al., 2007*), and single tones (*Miranda and Ullman, 2007*; *Omigie et al., 2013*; *Omigie et al., 2019*). These physiological markers enabled researchers to investigate the encoding of music expectations during development (*Koelsch et al., 2003*) as well as the impact of attention (*Loui et al., 2005*), short-term context (*Koelsch and Jentschke, 2008*), and musical experience (*Koelsch et al., 2002*) on music perception.

ERP studies (*Besson and Macar, 1987*; *Paller et al., 1992*; *Miranda and Ullman, 2007*; *Pearce et al., 2010b*; *Carrus et al., 2013*) often limit the range of expectations' violations strength that are tested to severe violations because of the need for repeated stimulus presentations. One issue with this approach is that notes eliciting strong violations could be considered by the listener as production mistakes (e.g., a pianist playing the wrong note), thus the corresponding cortical correlates may characterise only a limited aspect of the neural underpinnings of melodic perception. In fact, music sequences induce a wide range of violation strengths (*Pearce and Wiggins, 2012*) because the (valid) sequential events are not all equally likely (and hence not equally predictable), whether we consider note or chord sequences (*Temperley, 2008*) in both classical music (*Rohrmeier and Cross, 2008*) or more popular music genres (*Temperley and Clercq, 2013*). Here, we study the brain responses to continuous naturalistic musical stimuli, thus allowing us to explore the full range of expectations' strengths. This study seeks to determine whether the neural response reflects this range of strengths by regressing the musical information with it.

Music is a complex, multi-layered signal that have structures allowing predictions of a variety of properties. One layer concerns the notes sequences forming melodies, which is a key structural aspect of music across musical styles (*Reck, 1997*) and cultures (*Eerola, 2003*; *Pearce and Wiggins, 2006*). Pearce et al. designed a framework based on variable-order Markov models that learns statistics describing the temporal sequences in melodic sequences at various time-scales (IDyOM; *Pearce, 2005*; *Pearce and Wiggins, 2006*). This framework attempts to optimally predict the next note in a melodic sequence by combining predictions based on 1) long-term statistics learnt from a large corpus of western music and 2) short-term statistics from the previous notes of the current musical stream. In turn, this provides us with likelihood values for each note in a melody that have been shown to tightly mirror listeners' expectations (*Pearce, 2005*).

Here, we regressed these estimates with electrophysiological data to investigate the impact of expectations on auditory perception and cortical processing. We recorded scalp

electroencephalography (EEG; 20 subjects; about 1 hr and 15 min of data) signals and invasive electrocorticography (ECoG; three patients; about 25 min of data each) signals as participants listened to monophonic piano music from Bach that was generated from MIDI scores (see Materials and methods). According to the predictive coding theory (*Friston and Kiebel, 2009*; *Clark, 2013*), cortical signals partly reflect the mismatch between a participant's prediction and the actual sensory input. If this is the case, less expected musical notes should produce relatively stronger cortical responses. We expected individuals with musical expertise to internally generate predictions on a next note that more closely relate the ones of a specialized statistical model than non-musicians. We tested this hypothesis on our EEG dataset, which was recorded from both non-musicians and expert pianists, and investigated specific details of the cortical response using our ECoG dataset.

Our stimuli were regular musical streams, rather than artificially-constructed repeated patterns such as those usually utilized for the induction and detection of mismatched negativity (MMN) responses (*Garrido et al., 2009*; *Clark, 2013*; *Fishman, 2014*; *Lecaignard et al., 2015*; *Southwell and Chait, 2018*). Musical streams are imbued with melodic events that routinely violates listeners' expectation to some degree (*Pearce and Wiggins, 2012*; *Salimpoor et al., 2015*). Here we sought to test the hypothesis that the listeners' expectation violations produce cortical responses that change with the degree of the violations and that are measurable with EEG during naturalistic music listening. To establish the contribution of expectations (*Figure 1A*) to the neural responses, we used multivariate ridge regression (*Figure 1B*) to quantify how well the acoustic (**A**) factors (e.g., signal envelope and its derivative) and melodic expectation or surprise (**M**) factors (e.g., pitch and onset-timing) can predict the EEG and ECoG responses to music (*Crosse et al., 2016*). Since the prediction quality is considered to be an estimate of how strongly a stimulus property is encoded in the EEG data (*Di Liberto et al., 2015*; *Di Liberto et al., 2019*; *Brodbeck et al., 2018b*; *Somers et al., 2019*; *Verschueren et al., 2019*), and since cortical signals are assumed to be modulated by the various **A** and **M** factors above, we consequently expected the combination of both the acoustic and surprise features to predict the neural responses better than either set of features alone (*Figure 1C*). Validating these hypotheses would therefore provide physiological support for the melodic expectations generated according to the statistical learning model.

This work presents novel insights into the precise spatio-temporal dynamics of the integration of melodic expectations and sensory input during naturalistic musical listening. In turn, our results provide evidence of the neurophysiological validity of predictive statistical models of music structure. Crucially, we found distinct encoding of different melodic expectation features, (such as pitch and note onset) to the cortical responses to music.

## Results

Neural data were recorded from twenty healthy EEG participants and three ECoG epilepsy patients as they listened to monophonic excerpts of music from Bach sonatas and partitas that were synthesized with piano sound. The melodic expectation of each note was estimated from the musical score of the stimulus with IDyOM, the model for predictive statistical modelling of musical structure. Specifically, given a musical piece at time $t_0$, the model estimates the likelihood of having a note with a particular pitch at time $t_0$ given short-term information for $t < t_0$ and long-term information from a large corpus of Western music. Based on these estimates we calculated four measures (referred to as melodic features M, see Materials and methods for details) that capture distinct aspects of expectation and surprise at each new note within a melody: entropy of pitch ($H_p$), entropy of onset-time ($H_o$), surprise of pitch ($S_p$), and surprise of onset-time ($S_o$). The first two measures refer to the Shannon entropy at a particular position in a melody, before the musical note is observed. Intuitively, the entropy indicates the amount of uncertainty represented by the distribution of pitch and onset-time for the next note, where the most uncertain scenario is when all possible notes have equal likelihood and the least uncertain scenario is when the next note is known. The latter two measures refer to the *inverse* probability of occurrence of a particular note pitch or onset-time, with smaller values for more predictable transitions (see Materials and methods).

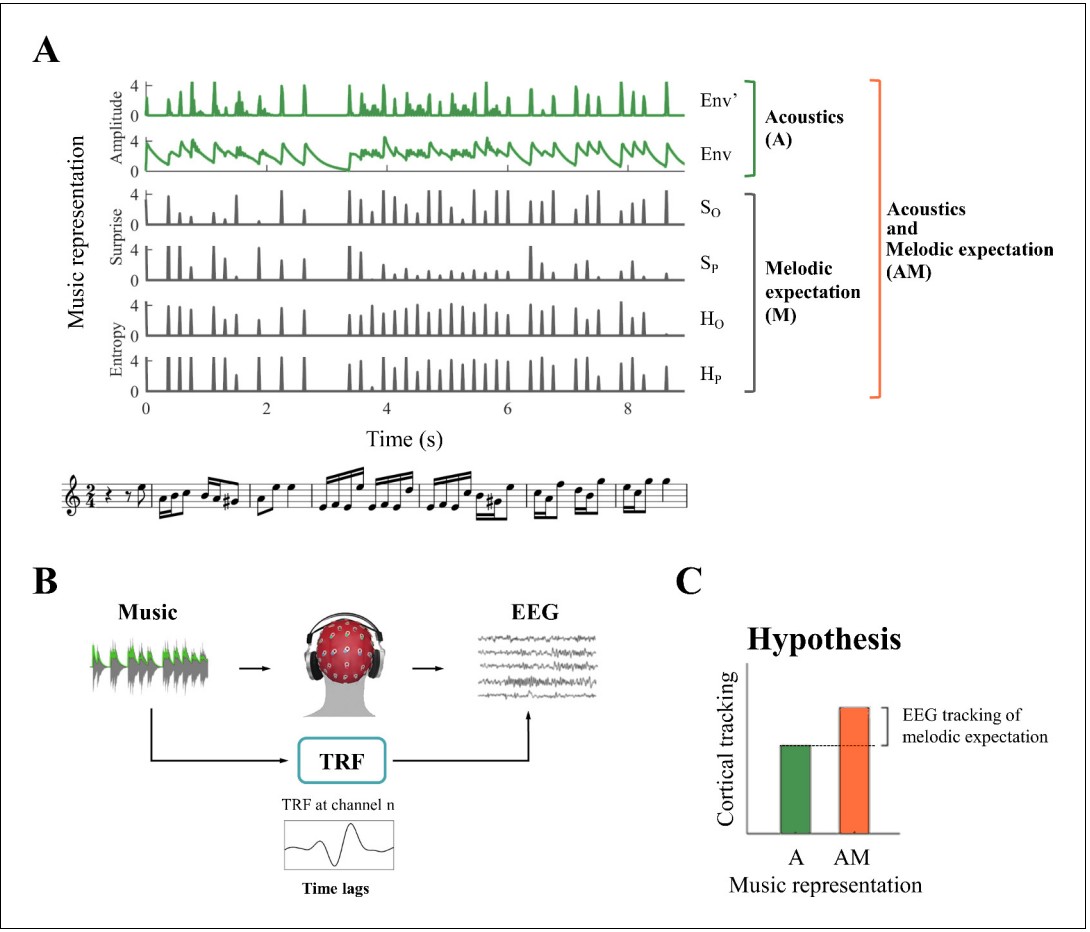

**Figure 1.** System identification framework for isolating neural correlates of melodic expectations. (**A**) Music score of a segment of auditory stimulus, with its corresponding features (from bottom to top): acoustic envelope (Env), half-way rectified first derivative of the envelope (Env'), and the four melodic expectation features: entropy of note-onset ($H_o$) and pitch ($H_p$), surprise of note-onset ($S_o$) and pitch ($S_p$). (**B**) Regularized linear regression models were fit to optimally describe the mapping from stimulus features (Env in the example) to each EEG and ECoG channel. This approach, called the Temporal Response Function (TRF), allows us to investigate the spatio-temporal dynamics of the linear model by studying the regression weights for different EEG and ECoG channels and time-latencies. (**C**) TRFs were used to predict EEG and ECoG signals on unseen data by using only acoustic features (A) and a combination of acoustic and melodic expectation features (AM). We hypothesised that cortical signals encode melodic expectations, therefore we expected larger EEG and ECoG predictions for the combined feature-set AM.

## Melodic expectation encoding in low-rate cortical signals

In all of the analyses and results below, we focused on the EEG and ECoG responses in the low-rate bands between 1 and 8 Hz, filtering out the remainder of the bands (see Materials and methods; note that inclusion of rates down to 0.1 Hz and up to 30 Hz did not alter any of the results that follow). Because of potential interactions between the responses to the succession of notes (which would complicate the interpretation of the ERPs time-locked to note onsets), we began by utilizing a linear modelling framework known as the temporal response function (TRF) (*Ding and Simon, 2012a*; *Crosse et al., 2016*) as depicted in *Figure 1B*. This approach 1) explicitly dissociates the effects of expectations from those due to changes in the acoustic envelope on the neural responses to music and 2) allows us to investigate neural responses to rapidly presented stimuli by accounting for the dependence among the sequences of input notes. Specifically, TRFs were derived by using ridge regression between suitably parameterized stimuli and their neural responses. These were then used to predict unseen EEG data (with leave-one-out cross-validation) based on either the acoustic properties alone (A predictions) or a combination of acoustics and melodic expectation

features (AM predictions). The predictive models included time-lagged versions of the stimulus accounting for delays between the stimulus and corresponding neural response. The time-lag window was limited to [0, 350] ms as longer latencies had little impact on the predictive power of the TRF model (see *Figure 2—figure supplement 1* and Materials and methods).

In *Figure 2* we illustrate the average (*Figure 2A*) and individual (*Figure 2B*) EEG prediction correlations for all subjects using either the A features alone (envelope and its derivative) or combined with the melodic expectation features, AM. The A correlations were significantly positive (p<0.05, permutation test) for all subjects but one (S20), confirming that neural responses to monophonic music track the stimulus envelopes (henceforth 'envelope tracking'). Crucially, AM correlations were significantly larger than A, implying that melodic expectations explained EEG variance that was not captured by acoustic information alone. Specifically, the average EEG prediction correlation over all electrodes was significantly larger for AM than for A both at the group level ($r_{AM} > r_A$: permutation test, $p < 10^{-6}$, $d = 1.64$; *Figure 2A*) and at the individual-subject level (16 out of 20 subjects, permutation test, $p < 0.05$; *Figure 2B*), and this difference was even larger when computed from selected single electrodes (e.g., Cz channel: $d = 1.80$). This supports the hypothesis that melodic expectation is directly reflected in the EEG responses to music.

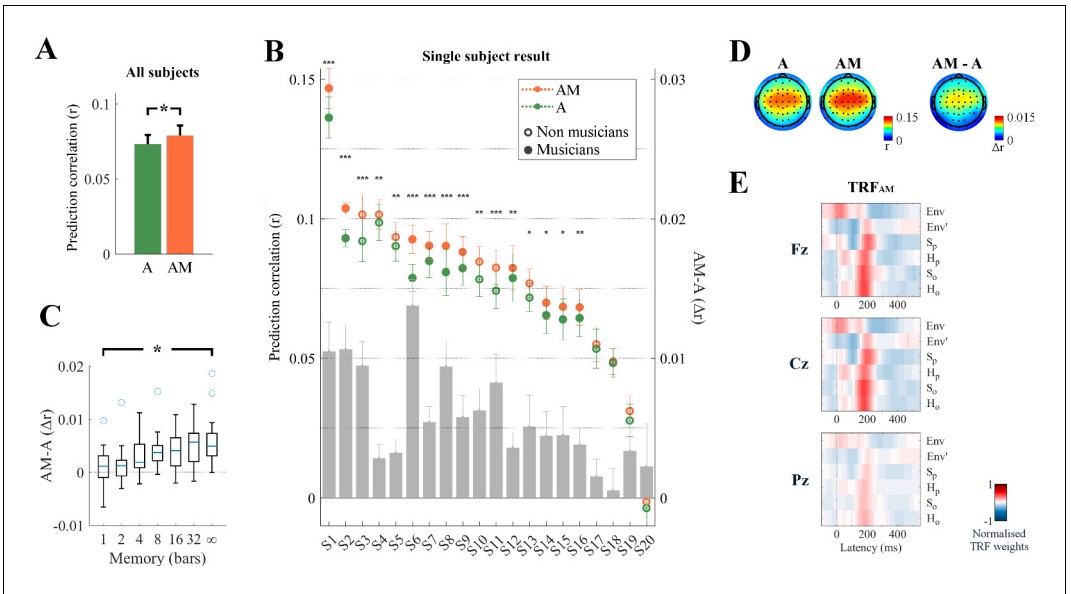

**Figure 2.** Low-rate (1–8 Hz) cortical signals reflect melodic expectations. Scalp EEG data were recorded while participants listened to monophonic music. Forward ridge regression models were fit to assess what features of the stimulus were encoded in the low-rate EEG signal. The link between music features and EEG was assessed by using these models to predict unseen EEG data. (A) Prediction correlations were greatest when the stimulus was described by a combination of acoustic information (A: envelope Env, and its half-way rectified first derivative Env') and melodic expectations (M: $S_P$, $S_O$, $H_P$, $H_O$). This effect of expectations was significant on the average prediction correlation across all 64 EEG electrodes (*p<10^{-6}). The error bars indicate the SEM across participants. (B) The enhancement due to melodic expectations emerged at the individual subject level. The gray bars indicate the predictive enhancement due to melodic expectation. Error bars indicate the SEM over trials (***p<0.001, **p<0.01, *p<0.05, permutation test). (C) Predictive enhancement due to melodic expectations (AM-A; y-axis) increased with the length of the local context (in bars; x-axis) used to estimate the expectation of each note (ANOVA: *p=0.0003). The boxplot shows the 25th and 75th percentiles, with the whiskers extending to the most extreme data-points that were not considered outliers. Circles indicate outliers. (D) The effect of melodic expectations ($r_{AM}-r_A$) emerged bilaterally on the same scalp areas that showed also envelope tracking. (E) Ridge regression weights for $TRF_{AM}$. Red and blue colors indicate positive and negative TRF components respectively.

The online version of this article includes the following figure supplement(s) for figure 2:

**Figure supplement 1.** The latency of the effect of melodic expectations on the low-rate EEG data (1–8 Hz) was assessed by measuring the loss in EEG prediction correlation when a given time-latency window is removed from the TRF fit.

While these results indicate that AM is a better descriptor of the signal than A, it should also be noted that TRF$_{AM}$ has higher dimensionality than TRF$_A$. Correlations are measured on unseen data, and thus should be immune to overfitting, nonetheless to verify that the better predictions for AM are not due simply to the higher degrees of freedom afforded by the addition of more M components, we assessed the performance of our TRF$_{AM}$ model using less elaborate M functions that nevertheless maintained the TRF$_{AM}$ with the same dimensionality and value distributions. Specifically, we built melodic expectation estimates by relying on progressively smaller amount of local context (or short-term memory) (*Figure 2C*), with M based on 1, 2, 4, 8, 16, and 32 musical bars, as opposed to the unbounded ('∞') estimates in our predictions of *Figure 2A*. In each of these cases, fitted TRFs of the same dimensionality performed better as the memory increased (*Figure 2C*; ANOVA: *F* (4.6,101.1) = 4.52, p=0.0003), indicating that the contribution of melodic expectation estimates to the prediction accuracy was not due to increased TRF dimensionality per se.

Despite the significant positive effects of melodic expectation on prediction correlations, we found no differences between the corresponding EEG topographical distributions for A, AM, or their difference AM-A (DISS$_{A,AM}$ = 0.017, p=0.33; DISS$_{A,AM-A}$ = 0.214, p=0.61; DISS$_{AM,AM-A}$ = 0.197, p=0.57; *Figure 2D*). By contrast, melodic expectations induced new *long temporal latencies* in the linear regression weights of the TRF$_{AM}$ model (*Figure 2E*), which were mostly centered around 200 ms compared to the 50 ms latency of the acoustic TRF$_A$ Env component (p<0.05, FDR correction; *Figure 2—figure supplement 1*).

## Melodic expectations modulate auditory responses in higher cortical areas

Since melodic expectations reflect regularities within a musical tone sequence at multiple time-scales that depend on the extent of knowledge and exposure of the subject listening to them, we hypothesized that neural signals correlated with the melodic properties of the music would be generated at higher hierarchical cortical levels than those strictly due to the acoustics (*Sammler et al., 2013*; *Bianco et al., 2016*; *Nourski et al., 2018*). EEG lacks the spatial resolution needed to test this hypothesis, but the test was possible in spatially localized ECoG recordings from three patients who had electrodes over the early primary auditory areas in the *anterior transverse temporal gyrus*, also called *Heschl Gyrus* (HG; patients 1 and 3), the belt regions along *planum temporale* (PT) and the *superior temporal gyrus* (STG), as well as the *supra-marginal gyrus* (SMG) in the parietal lobe (see *Supplementary file 1* for details on the channel locations and *Videos 1–3* and *Supplementary files 2–4* for a 3D view of the electrode placement). Although those regions are functionally heteroge-neous, our choice of anatomical division was motivated by both previous work indicating HG as the locus responsible for primary auditory processing (*Moerel et al., 2014*; *Nourski, 2017*), PT as an intermediary stage (*Griffiths and Warren, 2002*), and STG as a region involved the processing of high-level speech properties (*Chang et al., 2010*; *Mesgarani et al., 2014*). Both anatomical and functional studies measured a gradient change from the primary auditory processing in HG to the nonprimary areas in the lateral STG, and suggested a nonprimary role of PT (*Griffiths and Warren, 2002*; *Hickok and Saberi, 2012*), which is here considered as a higher cortical area. The inferior frontal gyrus (IFG) was expected to reflect melodic expectations as well, however we only had limited coverage in that cortical area. The subjects listened to the same monophonic music described earlier for the EEG experiments.

We first identified 21/241, 25/200, and 33/285 electrodes in Patients 1, 2, and three respectively that exhibited reliable auditory responses

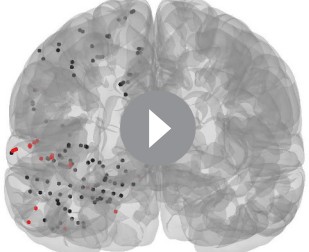

**Video 1.** Video showing the ECoG electrode placement in 3D for each of the three participants. Dots indicate ECoG channels. Red dots indicate channels that were responsive to the music input. The corresponding interactive Matlab 3D plots were also uploaded.
https://elifesciences.org/articles/51784#video1

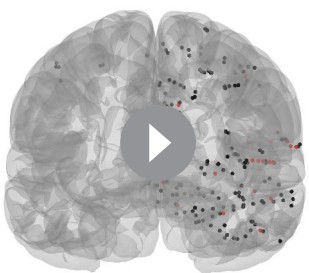

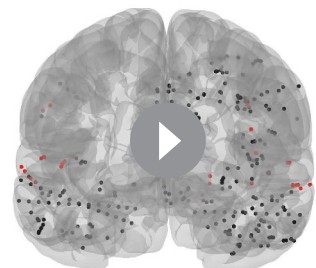

**Video 2.** Video showing the ECoG electrode placement in 3D for each of the three participants. Dots indicate ECoG channels. Red dots indicate channels that were responsive to the music input. The corresponding interactive Matlab 3D plots were also uploaded.

https://elifesciences.org/articles/51784#video2

**Video 3.** Video showing the ECoG electrode placement in 3D for each of the three participants. Dots indicate ECoG channels. Red dots indicate channels that were responsive to the music input. The corresponding interactive Matlab 3D plots were also uploaded.

https://elifesciences.org/articles/51784#video3

(stronger responses to monophonic music than to silence: Cohen's *d* > 0.5; *Figures 3C* and *4C*) in the form of either low-rate (1–8 Hz) local field potentials (similar bands to those in EEG analyses above), or power in the high-γ (70–150 Hz) field potentials which are thought to reflect local neuronal activity (*Miller et al., 2007*; see Materials and methods for details). A similar TRF analysis was conducted to identify responses that were sensitive to melodic expectations (*Figure 3A* and *Figure 3—figure supplement 1*). Envelope tracking was significant (permutation test on the ECoG prediction correlations over trials, p<0.05) in STG, PT, and HG channels. The predictive enhancement due to melodic expectations was small but significant (p<0.01, FDR-corrected) on several electrodes in PT in all patients, on one electrode in the *transverse temporal sulcus* (TTS), but not on the three HG electrodes in Patient 1, and was also significant on five bilateral HG electrodes in Patient 3. A Wilcoxon rank sum test indicated that the effect of expectations AM-A is larger in PT than HG (p=0.011; all electrodes in the two cortical areas from Patients 1 and 3 were combined, while Patient two was excluded as there was no coverage in HG). Similar effects as in PT were also measured in right parietal cortical areas (SMG and the *postcentral gyrus*) for Patient two but not for Patient 3. In addition, the effect of expectations seen in STG, PT, and HG was right lateralized in Patient 3 (p=0.038), while envelope tracking did not show a hemispheric bias (p=0.85).

*Figure 3B* depicts the TRF_AM weights for selected electrodes in SMG, PT, TTS, and HG. The TRF_AM weights for ECoG exhibited low-rate temporal patterns very similar to those measured with EEG. Specifically, strong correlations were found with the TRF_AM measured with EEG at Cz (*r* = 0.60, 0.50, and 0.45 in left PT, TTS, and HG respectively – e2, e6, and e9 from Patient 1; *r* = 0.61 and 0.80 in right PT and SMG – e4 and e10 from Patient two respectively). Overall, the TRF analysis of the ECoG recordings demonstrates that low-rate cortical responses to music encode melodic expectations. The strong similarities between ECoG responses and the EEG template obtained by averaging data from twenty participants (both musicians and non-musicians) suggests the possibility that the EEG melodic expectation results originate from temporal regions between PT and HG, and or parietal regions such as SMG. However, more direct evidence (with within-subject comparisons or source localization) are required to more confidently pinpoint the cortical origins of the EEG results.

ECoG recordings also allowed us to investigate more directly the link between local neuronal activity and melodic expectations, since these signals are available in the instantaneous power of the high-γ field potentials (*Crone et al., 2001*; *Edwards et al., 2009*; *Ray et al., 2008*; *Steinschneider et al., 2008*). Again, TRF analysis was used in *Figure 4* (and in *Figure 4—figure supplement 1*) to disentangle the contributions of envelope tracking and melodic expectations. As before, left HG (in Patients 1 and 3), left TTS (Patient 1), bilateral STG (Patients 2 and 3), bilateral PT

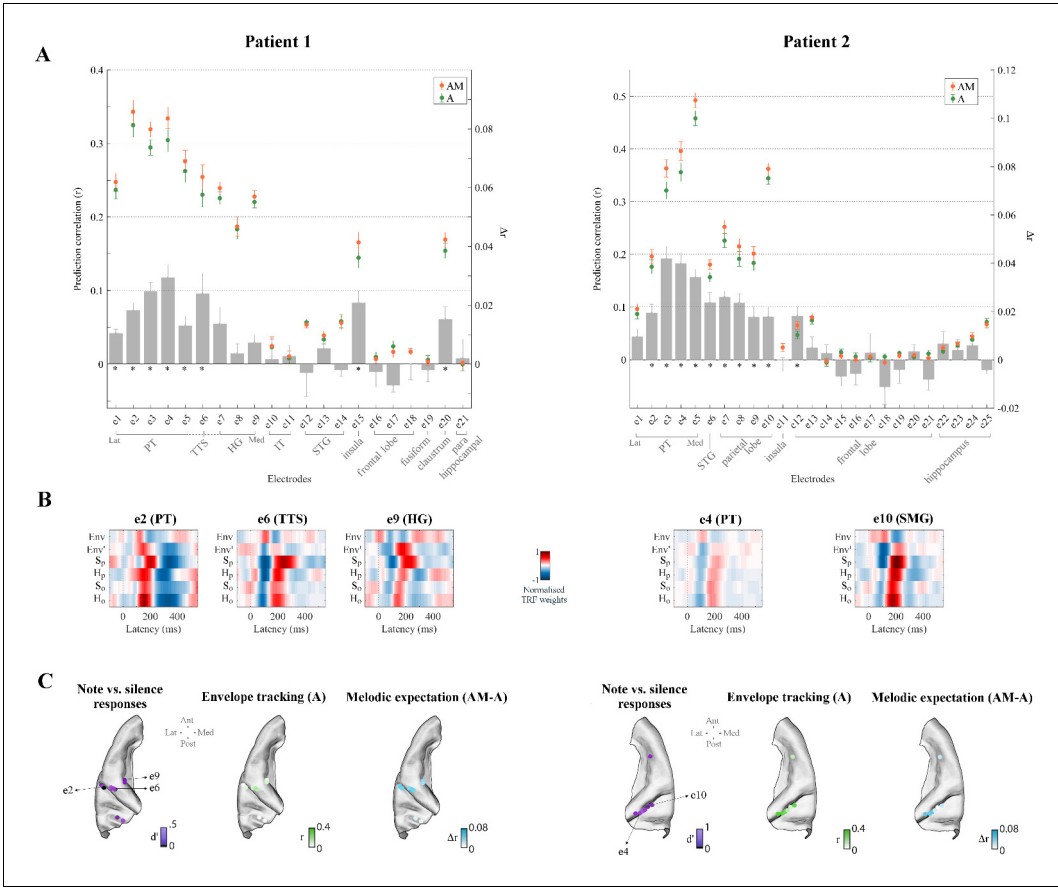

**Figure 3.** Low-rate (1–8 Hz) cortical signals in bilateral temporal cortex reflect melodic expectations. Electrocorticography (ECoG) data were recorded from three epilepsy patients undergoing brain surgery. Magnetic resonance imaging was used to localise the ECoG electrodes. Electrodes with stronger low-rate or high-γ (70–150 Hz) responses to monophonic music than to silence were selected (Cohen's $d > 0.5$). (A) ECoG prediction correlations for individual electrodes for A and AM. Electrodes within each group, as indicated by the gray square brackets, were sorted from lateral to medial cortical sites. The gray bars indicate the predictive enhancement due to melodic expectation ($r_{AM}-r_A$). Error bars indicate the SEM over trials (*p<0.01, FDR-corrected permutation test). (B) TRF weights for selected electrodes. For Patient 1, PT and TTS electrodes (e1-e5 and e6 respectively) exhibited large effects of musical expectations, while HG electrodes (e7-e9) had strong envelope tracking (A) but showed smaller effects of expectations that did not reach statistical significance. Patient two showed also strong envelope tracking and a significant effect of melodic expectations in the right temporal and parietal lobes (for example, the PT electrode e4 and the SMG electrode e10 respectively). (C) Low-rate (1–8 Hz) ECoG segments time-locked to note onsets were selected and compared with segments corresponding to silence. Colors in the first brain plot of each patient indicate the effect-size of the note vs. silence comparison (Cohen's $d > 0.5$). The second brain plot shows the EEG prediction correlations when using acoustic features only (A). The third brain plot depicts the increase in EEG predictions when including melodic expectation features (AM-A).

The online version of this article includes the following figure supplement(s) for figure 3:

**Figure supplement 1.** Bilateral electrocorticography (ECoG) results for Patient 3.

(Patients 1, 2, and 3), and right SMG (Patient two but not Patient 3) electrodes exhibited substantial envelope tracking. By contrast, the effect of expectations ($\Delta r = r_{AM}$ rA) was largest in PT, TTS, and in HG electrodes close to the junction between PT and HG, with a predictive enhancement up to ~50% (e.g., $\Delta r_{e6} = 0.09$ in Patient 1, which corresponds to a prediction enhancement $r_{AM,e6}/r_{A,e6}$ of 149%), in contrast to an enhancement of only 6% in the HG electrode with strongest envelope tracking (e9). Similar patterns emerged for Patient 3, with a predictive enhancement up to ~20% in PT and an enhancement of only 5% in the HG electrode with strongest envelope tracking (e9; but with stronger effects in other HG electrodes with weaker envelope tracking). Also, the temporal latencies in the

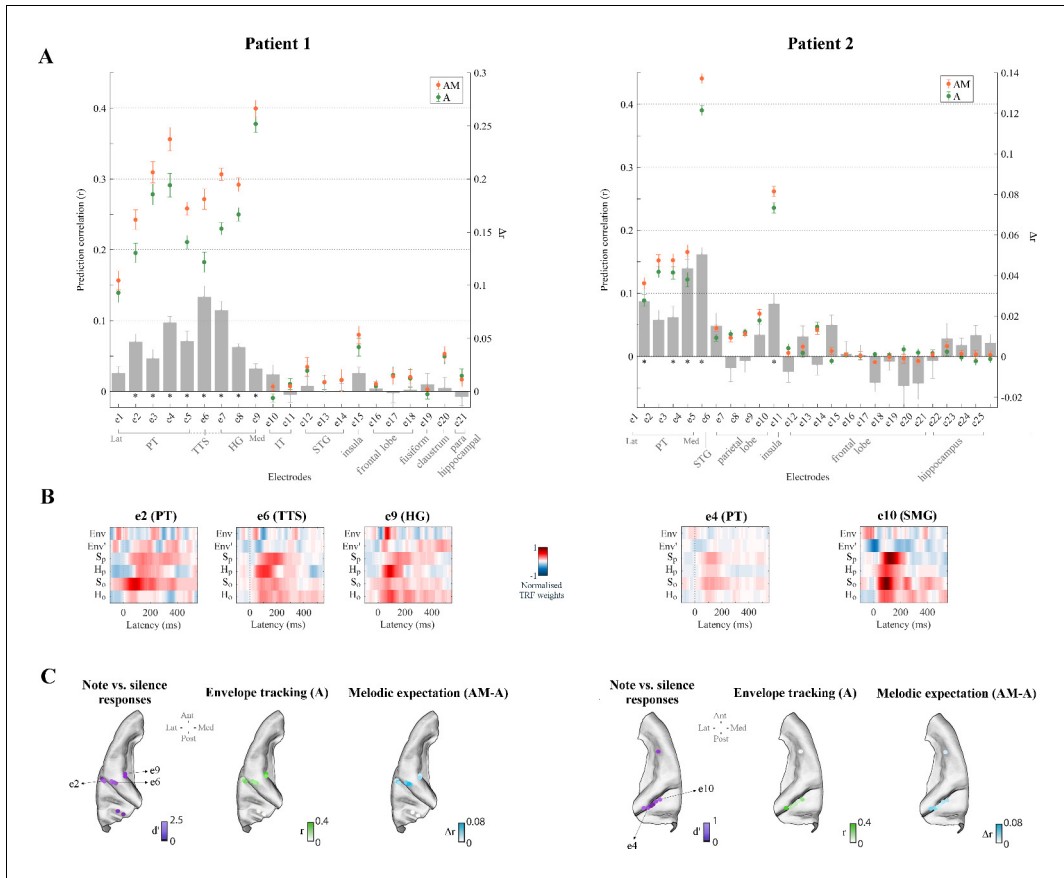

**Figure 4.** High-γ neural signals in bilateral temporal cortex reflect melodic expectations. Electrodes with stronger low-rate (1–8 Hz) or high-γ (70–150 Hz) responses to monophonic music than to silence were selected (Cohen's $d > 0.5$). (**A**) ECoG prediction correlations for individual electrodes for **A** and **AM**. Electrodes within each group, as indicated by the gray square brackets, were sorted from lateral to medial cortical sites. The gray bars indicate the predictive enhancement due to melodic expectation ($r_{AM}$-$r_A$). Error bars indicate the SEM over trials (*p<0.01, FDR-corrected permutation test). (**B**) Normalised TRF weights for selected electrodes (same electrodes as for *Figure 3*). For Patient 1, the HG electrode e9 showed the strongest envelope tracking and small effect of melodic expectations, while e6 in TTS exhibited the largest effect of expectations ($\Delta r_6 > \Delta r_9$, p=1.8e$^{-4}$, $d = 2.38$). For Patient 2, both e4 (PT) and e10 (SMG) electrodes showed strong envelope tracking and a significant effect of melodic expectations. (**C**) High-γ (70–150 Hz) ECoG segments time-locked to note onsets were selected and compared with segments corresponding to silence. Colors in the first brain plot of each patient indicate the effect-size of the note vs. silence comparison (Cohen's $d > 0.5$). The second brain plot shows the EEG prediction correlations when using acoustic features only (**A**). The third brain plot depicts the increase in EEG predictions when including melodic expectation features (**AM-A**).

The online version of this article includes the following figure supplement(s) for figure 4:

**Figure supplement 1.** Bilateral electrocorticography (ECoG) results for Patient 3.

TRF weights (*Figure 4B*) were rather different from what was previously seen for low-rate EEG and ECoG signals. In fact, the TRF$_A$ weights corresponding to the acoustic features exhibited sharp, short-latency dynamics while those of the melodic expectation features (TRF$_{AM}$) pointed to more temporally extended and strong neural responses.

## Explicit encoding of melodic expectations in the evoked-responses

So far, melodic effects were extracted in terms of the temporally extended analysis of the TRF, and indirectly validated through assessment of prediction accuracy. A more direct measure of these effects is possible by examining whether event-related potentials (ERPs) time-locked to note onsets are specifically modulated by melodic expectations, that is *beyond* what is expected from the

acoustic features of the stimuli. For instance, here we specifically demonstrate that the cortical responses evoked by tones of identical envelope can produce significantly different responses that are modulated proportionately to the melodic values of the tones. To do so, we selected notes with equal acoustic envelopes corresponding to the median peak envelope amplitude (25% of all notes), but had large disparity in their surprise values $S_p$ according to the IDyOM model, namely the top 20% and bottom 20% (*Figure 5*, purple and pink respectively). As illustrated in *Figure 5*, the two groups (purple and pink curves) had identical average signal envelopes (*Figure 5A*), but displayed significantly disparate EEG responses (*Figure 5B*), with significantly larger responses to the notes with the high pitch surprise (purple >pink; p<0.05 at the N1 and P2 peaks, permutation test;

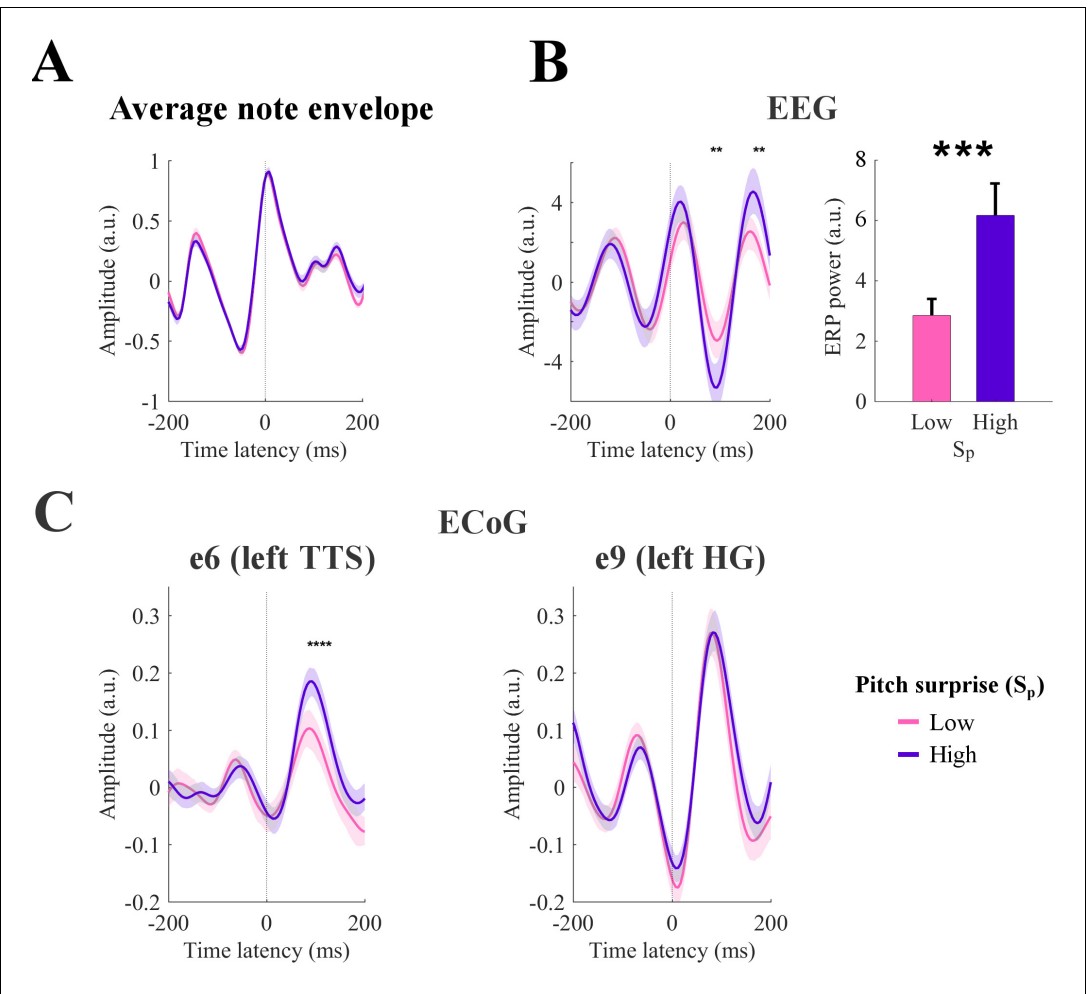

**Figure 5.** Event-related potentials (ERP) analysis. (**A**) Notes with equal peak envelope were selected (median envelope amplitude across all notes with a tolerance of ±5%). Together, the selected elements were 25% of all notes. Notes were grouped according to the corresponding pitch surprise values ($S_p$). The figure shows the average sound envelope for the 20% notes with lowest and highest surprise values. Shaded areas indicate the 95% confidence interval. (**B**) Low-rate EEG signals time-locked to note-onset were selected for high and low $S_p$ values. ERPs for channel Cz are shown on the left. Shaded areas indicate the 95% confidence interval (across subjects). Stars indicate significant differences between ERPs for high and low surprise for a given note onset-EEG latency (permutation test, p<0.05, FDR-corrected). The right panel shows the total ERP power for the latencies from 0 to 200 ms (*p<0.001, permutation test). Error-bars indicate the SEM across subjects. (**C**) ERPs for high-γ ECoG data from left TTS and HG (Patient 1). Stars indicate significant differences between ERPs for high and low surprise for a given note onset-EEG latency (permutation test, p<0.05, FDR-corrected). Shaded areas indicate the 95% confidence interval (across individual trials, that is responses to single notes).

The online version of this article includes the following figure supplement(s) for figure 5:

**Figure supplement 1.** Event-related potentials (ERP) analysis.

p=0.001 on the power of the average ERP across all channels for latencies between 0 and 200 ms). A similar effect emerged for $H_p$ and $H_o$ (average power ERP within 0–200 ms, high surprise >low surprise with p=0.0425 and p=0.006 for $H_p$ and $H_o$ respectively; *not shown*), while no effect was measured for $S_o$ (p=0.8764). Note that the ERPs showed large responses at pre-stimulus latencies (before zero latency). This is due to the temporal regularities that are intrinsic in music, which results in large average envelope before the note of interest (see *Figure 5A*). In fact, limiting the ERP calculation to musical events with preceding inter-note-interval longer than 200 ms eliminated such pre-stimulus responses from the ERPs (*not shown*). However, this selection procedure reduced the number of EEG epochs, and thus our choice to include short inter-note-interval in the analysis in *Figure 5*.

Similar analyses for both low-rate and high-γ ECoG data revealed that ERP responses in TTS electrodes to musical notes with *equal* envelopes were modulated in proportion to the $S_p$ (*Figure 5C*) stats; see *Figure 5—figure supplement 1*). This effect of melodic surprise was absent in the electrode with strongest envelope tracking e9 in Patient 1 (in the left HG; *Figure 5C*). These results are consistent with previous findings on melodic expectations (*Omigie et al., 2013*; *Omigie et al., 2019*) and in line with the hypothesis that higher stimulus expectation can reduce auditory responses (*Todorovic et al., 2011*; *Todorovic and de Lange, 2012*). Furthermore, this result complements the TRF analysis by confirming that the effect of melodic expectations on the cortical responses can be disentangled from changes in the amplitude of the stimulus envelope. It should be emphasized, however, that compared to the TRF approach, this analysis may in many cases suffer from the potential of interactions between the responses to the sequence of notes, for example if the internote interval is shorter than the duration of the neural response of interest. It also cannot isolate among the interactions and modulations due to the various melodic expectation features. Nevertheless, the validity of these results is confirmed by the parallel TRF findings in *Figures 2–4*, that the encoding of melodic expectations in the cortical responses is different from responses due to stimulus acoustics.

## Pitch and onset-time induce distinct musical expectations

So far, we have parameterized melodic expectations in terms of surprise and entropy features, each for pitch and note-onsets. Surprise and entropy were expected to interact as they convey complementary information (*Cheung et al., 2019*; *Gold et al., 2019*). Entropy provides information on the uncertainty of the prediction of the next note before observing the event, thus it describes the overall probability distribution. Surprise depends on that same distribution but is specific to the observed event. For this reason, we expected the responses to entropy and surprise to be dissociable in their temporal dynamics. This hypothesis was tested in our EEG data by measuring the contrast in the $TRF_{AM}$ weights for surprise *versus* entropy (*Figure 6A*, top; weights were averaged as follows: $(S_p+S_o)/2$ vs. $(H_p+H_o)/2$). The results showed that responses with latencies up to 350 ms were significantly dominated by both surprise and entropy in alternation (p<0.05, permutation test, FDR-corrected).

A second analysis was conducted to test the relative contribution of pitch and onset-time expectations to the $TRF_{AM}$ model. As previous studies suggested a dissociation between pitch and sound onset processing (*Schönwiesner and Zatorre, 2008*; *Coffey et al., 2017*), we expected similar differences in the processing of their expectations in early auditory cortical regions. We tested for such a dissociation in our EEG data by measuring the contrast in the $TRF_{AM}$ weights for pitch *versus* onset time (*Figure 6A*, bottom; $(S_p+H_p)/2$ vs. $(S_o+H_o)/2$). Note-onset dominant responses emerged only up to 200 ms, while pitch dominant responses persisted for much longer latencies up to 400 ms. The latency differences for pitch and note-onset TRFs suggests a certain level of dissociation between pitch and onset-time expectations.

Our results indicate that brain responses to music are modulated by melodic expectations, an effect that was explicitly accounted for by including **M** in the TRF mapping, and are in line with the hypothesis that more surprising notes elicit larger auditory responses (*Todorovic et al., 2011*; *Todorovic and de Lange, 2012*; *Chennu et al., 2013*; *Auksztulewicz and Friston, 2016*). Accordingly, musical pieces with higher mean surprise values were expected to elicit EEG and ECoG responses with higher SNR, thus producing larger prediction correlation scores. To test this hypothesis, we calculated the mean scores for each expectation feature (*Figure 6B*) and measured their correlation with the envelope tracking (*Figure 6—figure supplement 1*). Significant Spearman correlations were measured between the average $S_o$ of a piece and the neural signal prediction

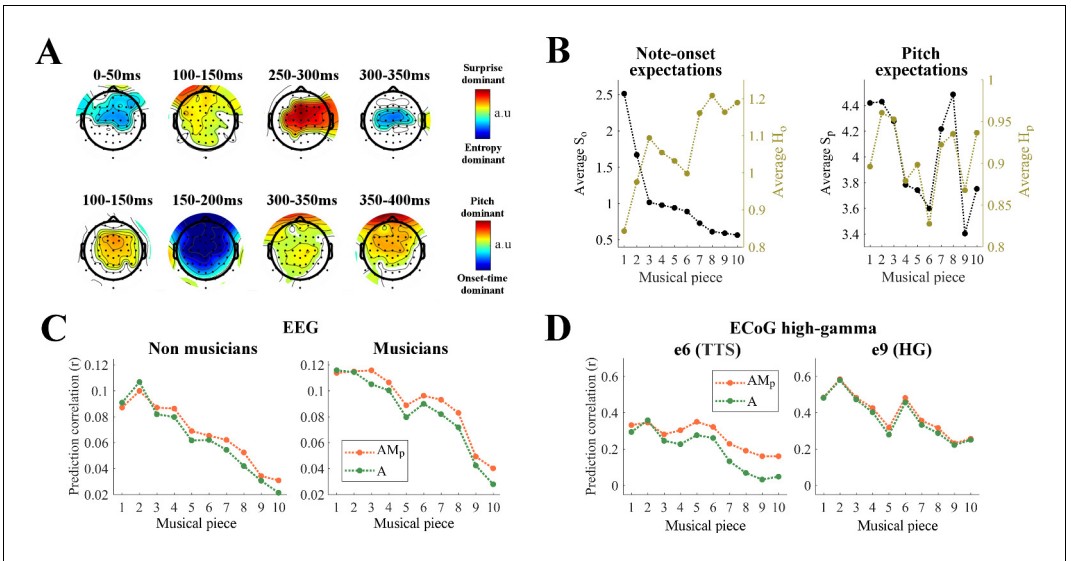

**Figure 6.** Distinct cortical encoding of pitch and note onset-time during naturalistic music listening. (**A**) Contrasts at each EEG channel of the TRF weights for surprise vs. entropy (top) and pitch vs. onset-time (bottom) in $TRF_{AM}$. Colors indicate significant differences ($p<0.05$, permutation test, FDR-corrected) (**B**) Average surprise and entropy of note-onsets ($S_o$ and $H_o$) and of pitch ($S_p$ and $H_p$) for each musical piece. Musical pieces were sorted based on $S_o$, where lower average $S_o$ indicates musical pieces with more predictable tempo. (**C**) Cortical tracking of music changes with overall surprise of note onset-time within a musical piece. Single-trial EEG prediction result (average across all channels) for musicians ($N_m = 10$) and non-musicians ($N_n = 10$). Trials were sorted as in panel B. (**D**) Single-trial ECoG prediction correlations for the surgery Patient one for two electrodes of interest.
The online version of this article includes the following figure supplement(s) for figure 6:

**Figure supplement 1.** Scatter plots indicating the correlation between EEG prediction correlation using the acoustic regressors A for each musical piece and the average expectation score (Sp, Hp, So, or Ho) for all notes of the corresponding piece.

---

correlations for EEG ($r = 0.98$, $p<0.001$ for non-musicians; $r = 0.96$, $p<0.001$ for musicians; *Figure 6C*) and high-$\gamma$ ECoG data ($r = 0.88$, $p=0.002$ for e6 in the left TTS of Patient 1; $r = 0.88$, $p=0.002$ for e9 in the left HG of Patient 1; *Figure 6D*). These effects were specific to onset-time surprise. In fact, Spearman correlations of comparable magnitude emerged with $-H_o$, while no significant correlations were measured for $S_p$ and $H_p$ for these pieces (*Figure 6B*, *Figure 4—figure supplement 1*). *Figure 6C and D* also illustrates the prediction correlations for $AM_p$, showing that small (nearly zero) envelope tracking due to small average $S_o$ does not hamper the encoding of pitch expectations on the same ECoG electrode (see *Figure 6D* left), thus further highlighting the dissociation of processes underlying expectation of pitch and onset-time.

## Effect of musical expertise on the encoding of melodic expectations

We were also able to shed light on the effect of musical expertise on the encoding of melodic expectations. Specifically, by design, half of the EEG participants had no musical training, while the others were expert pianists that studied for at least ten years (*Figure 2*). In *Figure 7A* we show a comparison between the two EEG groups. A cluster statistics indicated that melodic expectation was larger for musicians than non-musicians for frontal EEG channels (*Figure 7B*; see Di Liberto et al. in press, for comparisons that are specific to music envelope tracking). Note that subjective reporting indicated no significant effect of musical training on the familiarity with the musical pieces (see Materials and methods).

## Discussion

Musical perception is strongly influenced by expectations (*Bar et al., 2006*; *Huron, 2006*; *Kok et al., 2012*; *Pearce, 2018*; *Henin et al., 2019*). Violation of these expectations elicits distinct

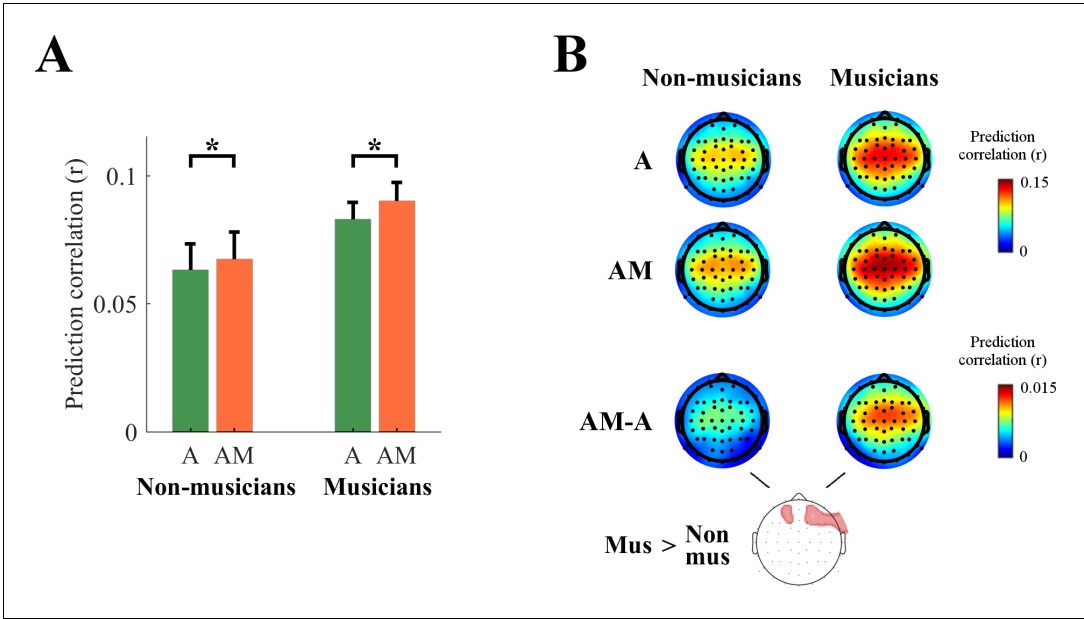

**Figure 7.** Effect of musical expertise on the low-rate encoding of melodic expectations. (**A**) EEG prediction correlations (average across all scalp channels) for musicians and non-musicians (*p<10⁻⁵). The error bars indicate the SEM across participants. (**B**) EEG prediction correlations for individual scalp electrodes (red color indicates stronger prediction correlations) for **A**, **AM**, and **AM-A**. A cluster statistics found a stronger effect of melodic expectations for musicians.

neural signatures that may underlie the emotional and intellectual engagement with music (*Zatorre and Salimpoor, 2013*; *Cheung et al., 2019*; *Gold et al., 2019*). Here, we exploited such neural signatures of melodic expectations during listening to Bach monophonic pieces to demonstrate that cortical responses encode explicitly subtle changes in note predictability during naturalistic music listening. In doing so, we 1) demonstrated a novel methodology to assess the cortical encoding of melodic expectations that is effective at the individual subject level with non-invasive EEG signals recorded during passive music listening, 2) provided detailed insights into the spatial selectivity and temporal properties of that encoding in ECoG recordings, and 3) physiologically evaluated the statistical learning framework of melodic expectation, exemplified by models such as IDyOM (*Pearce, 2005*).

Our findings have several implications for current theories and understanding of sensory perception in general, and musical cognition in particular. First, we show that the neural responses are consistent with the statistical learning theoretical frameworks in that response dynamics are proportional to the predictability of several signal attributes, including pitch and onset-timing. For musical sequences, additional attributes, such as timbre, loudness, and complex patterns of harmony are likely to be relevant, although we did not investigate them here. These properties may all contribute in tandem to music perception, or sometimes in a dissociated manner, as we demonstrated for the relative contribution of pitch and onset-timing (see also *Coffey et al., 2017*). Second, melodic expectations attributes are encoded in both low-rate and high-γ responses of higher cortical regions (e.g., bilateral PT in humans). The effect of expectations extended to the junction between the PT with HG (TTS) and to neighbouring electrodes in HG, an early auditory cortical area that favors the accurate representation of the acoustic properties of the music. In addition, neural signals in some SMG electrodes showed similar encoding of melodic entropy and surprise, a result that well aligns with previous findings suggesting a role of the left SMG in short-term pitch memory (*Vines et al., 2006*; *Schaal et al., 2015*; *Schaal et al., 2017*). Third, we have demonstrated a methodology to objectively assess the cortical encoding of melodic expectations using non-invasive neural recordings and passive listening of ecologically valid stimuli. This finding combined with recent evidence on the precise link between pleasure of music listening and the particular combinations of

surprise and entropy of the stimuli (*Cheung et al., 2019*; *Gold et al., 2019*) opens new avenues for investigation of the neurophysiological bases of music perception and its link with emotions (*Salimpoor et al., 2009*; *Salimpoor et al., 2013*; *Shany et al., 2019*).

This work builds upon previous ERP findings on tone repetitions (*Todorovic et al., 2011*) and simple note sequences (*Omigie et al., 2013*; *Omigie et al., 2019*), but circumvents one of the main issues tied to the ERP approach, namely that it does not allow in general to isolate responses to continuous or rapidly presented stimuli. While such limitation may be overcome for stimuli with particular statistics, such as speech (*Khalighinejad et al., 2017*), the temporal regularities inherent in music hamper the ability to dissociate the late ERP components in response to a note from the early responses to subsequent ones. Instead, the TRF framework is more resistant to this issue as it assumes that the transformation of a stimulus into the corresponding brain responses can be captured by a linear time-invariant system. Although the human brain is neither linear nor time-invariant, these assumptions can be reasonable in certain cases, and this approach was shown to be effective in previous studies with stimuli that were either discrete and rapidly presented, or continuous (*Lalor et al., 2006*; *Ding and Simon, 2012b*; *Di Liberto et al., 2015*; *Crosse et al., 2016*; *Fiedler et al., 2017*; *Brodbeck et al., 2018a*; *Broderick et al., 2018*; *Vanthornhout et al., 2018*; *Wong et al., 2018*). Using this approach, we have demonstrated how information on melodic expectations, as well as levels and fidelity of envelope tracking, can be extracted from both EEG and ECoG recordings. There is ample evidence that melodic expectations of various origins modulate responses to stimulus acoustics, possibly increasing the magnitude of those responses according to the degree of violation of the expectations. Such a modulation was captured by our multivariate TRF analysis in *Figure 2E*, which emerged as a positive expectation component at latencies between 150 and 250 ms, the same latencies corresponding to ERP components elicited by musical violations such as the ERAN (*Koelsch et al., 2002*). The present findings go beyond previous work by investigating the encoding of melodic expectations in valid sequences with high temporal (EEG and ECoG) and spatial (ECoG) resolution. However, similar to ERP analyses, the TRF approach is most effective in capturing signals that are precisely time-locked to note onset. Therefore, further studies with different methodologies are required to assess melodic expectation encoding on cortical sites with low or no time-locked responses to musical notes, where the TRF methodology was less effective. Despite this limitation, the TRF approach allowed us to extract objective indices of melodic expectations encoding that were derived by both explicit (**AM-M**; *Figures 2*, *3*, *4* and *7*) and implicit (**A**-only; *Figure 6*) analyses.

Furthermore, since subjects' expectations modulate musical responses differently depending on their cultural experience and musical exposure, it is expected that musical expertise may significantly enhance these modulations and hence reveal stronger encoding of melodic expectations. The evidence we present in this study (*Figure 7*) is in line with this view and, although preliminary, the finding is consistent with previous neuroimaging results showing effects of musical training on the brain responses to music in both children and adults (e.g. *Jentschke and Koelsch, 2009*; *Oechslin et al., 2013*). The results in *Figure 7* leave open a key question: Do musicians in general encode melodic expectations better (more strongly or accurately), or is the estimate of the IDyOM model more in tune with that of musicians' predictions than non-musicians'? Our results also suggest the possibility of a right hemispheric bias in the processing of melodic expectations, and the separate analysis of low-rate and high-γ neural signals seems crucial to investigate such an effect (*Figure 7B*, *Figure 3—figure supplement 1*, and *Figure 4—figure supplement 1*). Other possibilities need to be controlled for in future studies with larger sample sizes and more types of stimuli, for example whether the experiment may have been more engaging for musically trained participants, which would explain their stronger envelope tracking! It will also be important to assess the effect of musical expertise on the optimal amount of memory for estimating melodic expectations.

The present study demonstrates that rich predictive models can be combined with neural recordings to disentangle the processing of distinct properties of complex sensory inputs. Previous work demonstrated that this approach is effective in other domains, such as in the study of natural speech perception, where neural responses at the level of acoustics, phonemes, phonotactics, and semantics were measured (*Di Liberto et al., 2015*; *Di Liberto et al., 2019*; *Brodbeck et al., 2018c*; *Broderick et al., 2018*). The ability to investigate multiple domains with a same framework may contribute to the search for fundamental shared neural mechanisms (*Patel, 2003*; *Fitch and Martins, 2014*). For instance, the hypothesis of shared resources between language and music has been

supported by evidence of overlapping brain responses for processing music and language structures (*Maess et al., 2001*). However, it is unclear whether these effects reflect domain-specific computations or domain-general functions such as working memory or cognitive control (*Rogalsky et al., 2011*; *Fitch and Martins, 2014*), or what properties in the stimuli provide the most parsimonious comparison across these domains (*Heffner and Slevc, 2015*). Indeed, there is a fundamental difference in the use of predictions in the two domains. In speech, expectations are important to successfully understand the meaning of a sentence (e.g., phonemic restoration, priming; *Leonard et al., 2016*; *Norris et al., 2016*), especially in noisy, multi-talker environments (*McGettigan et al., 2012*; *Strauß et al., 2013*). In music, expectations may have a stronger link to emotions and musical engagement (*Dunsby, 2014*; *Salimpoor et al., 2015*).

Our results on melodic expectations exhibited spatio-temporal patterns that are different from those for the typical ERPs for syntactic and semantic violations in the case of natural speech perception, (N400, P600; for example *Osterhout and Holcomb, 1995*; *Kutas and Federmeier, 2011*; *Borovsky et al., 2012*). Furthermore, previous work on phonotactic- and semantic-level expectations that used system identification methods as in the present study (*Brodbeck et al., 2018c*; *Broderick et al., 2018*; *Di Liberto et al., 2019*) exhibited strong TRF centro-parietal components at latencies around 300–500 ms, which were absent in the responses to melodic surprise and entropy. Instead, our findings align nicely with previous results for syntax surprisal responses based on recurrent neural network models of language structure (*Hale et al., 2018*). This calls for further investigations with explicit within-subject comparisons, with the present findings providing a key starting point to tackle these questions. One factor that may be crucial in this investigation is the ability to exploit different expectation models to disentangle different contributors to expectations, such as statistical learning rule-based processing (*Morgan et al., 2019*). In fact, although statistical learning (IDyOM) was shown to have a prominent role on melodic expectations based on behavioral data, an independent contribution of a rule-like music-theoretically motivated approach was found (Temperley Probabilistic Model of Melody Perception; *Temperley, 2008*). In this sense, it is possible that the melodic expectation signals presented here only partly represent music responses.

This study presented novel detailed insights on the impact of melodic expectations on the neural processes underlying music perception and informed us on the physiological validity of models of melodic expectations based on Markov chains. In the process, we introduced the first solution to investigate the neural underpinnings of music perception in ecologically-valid listening conditions. As a result, this work constitutes a platform where research in cognitive neuroscience and musicology meet and can inform each other.

## Materials and methods

### EEG data acquisition and preprocessing

Twenty healthy subjects (10 male, aged between 23 and 42, M = 29) participated in the EEG experiment. Ten of them were highly trained musicians with a degree in music and at least ten years of experience, while the other participants had no musical background. Each subject reported no history of hearing impairment or neurological disorder, provided written informed consent, and was paid for their participation. The study was undertaken in accordance with the Declaration of Helsinki and was approved by the CERES committee of Paris Descartes University (CERES 2013–11). The experiment was carried out in a single session for each participant. EEG data were recorded from 64 electrode positions, digitized at 512 Hz using a BioSemi Active Two system. Audio stimuli were presented at a sampling rate of 44,100 Hz using Sennheiser HD650 headphones and Presentation software (http://www.neurobs.com). Testing was carried out at École Normale Supérieure, in a dark room, and subjects were instructed to maintain visual fixation on a crosshair centered on the screen, and to minimize motor activities while music was presented. A preliminary analysis was conducted on part of this EEG dataset in a separate study that compared EEG tracking in musicians and non-musicians (*Di Liberto et al., 2020*).

Neural data were analysed offline using MATLAB software (The Mathworks Inc). EEG signals were digitally filtered between 1 and 8 Hz using a Butterworth zero-phase filter (low- and high-pass filters both with order two and implemented with the function *filtfilt*), and down-sampled to 64 Hz. Results were also reproduced with high-pass filters down to 0.1 Hz and low-pass filters up to 30 Hz. EEG

channels with a variance exceeding three times that of the surrounding ones were replaced by an estimate calculated using spherical spline interpolation. All channels were then re-referenced to the average of the two mastoid channels with the goal of maximising the EEG responses to the auditory stimuli (*Luck, 2005*).

## ECoG data acquisition and preprocessing

We recorded cortical activity from three adult human patients (one male) implanted with stereotactic EEG electrodes at the Northwell Health University Hospital as part of their clinical evaluation for epilepsy surgery. The research protocol was approved and monitored by the institutional review board at the Feinstein Institute for Medical Research (07–125), and written informed consent of the patients was obtained before surgery. The first patient (P1) was a highly trained musician with about twenty years of experience; the second patient (P2) had no musical background; and the third patient (P3) studied clarinet for 8 years while in secondary school and had not played for 25 years. As a part of their clinical diagnosis of epileptic focus, P1 was implanted with a total of 241 electrodes in the left hemisphere, P2 with 200 electrodes in the right hemisphere, and P3 with 285 electrodes in both left and right hemispheres. Patients had self-reported normal hearing. Electrocorticography signals with sampling rate of 3000 Hz were recorded with a multichannel amplifier connected to a digital signal processor (Tucker-Davis Technologies). All data were montaged again to common average reference (*Crone et al., 2001*).

Channel positions were mapped to brain anatomy using registration of the postimplantation computed tomography (CT) to the preimplantation MRI via the postoperative MRI (*Groppe et al., 2017*). The CT was first coregistered with the postimplantation structural MRI and, subsequently, with the preimplantation MRI. The coregistration was performed by means of the automated procedure FSL's FLIRT (*Mehta and Klein, 2010*). Channels were assigned to anatomical areas according to the Destrieux atlas (*Destrieux et al., 2010*) and confirmed by expert inspection blinded to the results of this study.

Neural responses were transformed using Hilbert transform to extract the high-$\gamma$ band (70–150 Hz) for analysis (*Edwards et al., 2009*). This signal is known to correlate with neural spiking activity (*Ray et al., 2008*; *Steinschneider et al., 2008*) and was shown to reliably reflect auditory responses (*Kubanek et al., 2013*; *Mesgarani et al., 2014*). Secondly, low-rate responses were extracted from the raw unfiltered data by digitally filtering between 1 and 8 Hz using a Butterworth zero-phase filter (low- and high-pass filters both with order two and implemented with the function *filtfilt*). Both high-$\gamma$ and low-rate signals were then down-sampled to 100 Hz.

Music-responsive sites (i.e. with significant electrical potentials time-locked to note onset) were determined by comparing portion of ECoG responses to music with signals recorded during the pre-stimulus silence. 25 chunks of data, each with duration 200 ms, were selected for each of the two conditions and Cohen's $d$ effect-size was calculated to quantify the effect of a monophonic music stimulus on the ECoG data. Electrodes with $d > 0.5$ (medium effect-size) were marked as music-responsive (21, 25, and 34 electrodes for patients 1, 2, and three respectively).

## Stimuli and procedure

Monophonic MIDI versions of ten musical pieces from Bach's monodic instrumental corpus were partitioned into short snippets of approximately 150 s. The selected melodies were originally extracted from violin (partita BWV 1001, presto; BWV 1002, allemande; BWV 1004, allemande and gigue; BWV 1006, loure and gavotte) and flute (partita BWV1013 allemande, corrente, sarabande, and bourrée angloise) scores and were synthesised by using piano sounds with MuseScore two software (MuseScore BVBA), each played with a fixed rate (between 47 and 140 bpm). This was done in order to reduce familiarity for the expert pianist participants while enhancing their neural response by using their preferred instrument timbre (*Pantev et al., 2001*). Each 150 s piece, corresponding to an EEG/ECoG trial, was presented three times throughout the experiment, adding up to 30 trials that were presented in a random order. At the end of each trial, participants were asked to report on their familiarity to the piece (from 1: unknown; to 7: know the piece very well). This rating could take into account both their familiarity with the piece at its first occurrence in the experiment, as well as the build-up of familiarity across repetitions. Behavioural results confirmed that participants reported repeated pieces as more familiar (paired t-test on the average familiarity ratings for all participants

across repetitions: $rep_2 > rep_1$, p=$6.9\times10^{-6}$; $rep_3 > rep_2$, p=0.003, Bonferroni correction). No significant difference emerged between musicians and non-musicians on this account (two-sample $t$-test, p=0.07, 0.16, 0.19 for repetitions 1, 2, and three respectively). EEG participants undertook the entire experiment (30 trials: ten stimuli repeated three times), ECoG patients were presented with 10 trials (ten stimuli, played with random order).

## IDyOM

The Information Dynamics of Music model (IDyOM; *Pearce, 2005*) is a framework based on variable-order hidden Markov models. Given a note sequence of a melody, the probability distribution over every possible note continuation is estimated for every $n$-gram context up to a given length $k$ (model order). The distributions for the various orders were combined according to an entropy-based weighting function (IDyOM; *Pearce, 2005*), Section 6.2). Here, we used an unbounded implementation of IDyOM that builds $n$-grams using contexts up to the size of each musical piece. In addition, predictions were the result of a combination of long- and short-term models (LTM and STM respectively), which yields better estimates than either LTM or STM alone. The LTM was the result of a pre-training on a large corpus of Western music that did not include the stimuli presented during the EEG experiment, thus simulating the statistical knowledge of a listener that was implicitly acquired after a life-time of exposure to music. The STM, on the other hand, is constructed online for each individual musical piece that was used in the EEG experiment.

Our choice of IDyOM was motivated by the empirical support that Markov model-based frameworks received as a model of human melodic expectation (*Pearce and Wiggins, 2006*; *Pearce et al., 2010a*; *Omigie et al., 2013*; *Quiroga-Martinez et al., 2019a*). Specifically, among other evidence, previous work has indicated that it predicts human ratings of uncertainty during music listening (*Hansen and Pearce, 2014*; *Moldwin et al., 2017*; *Bianco et al., 2020*).

## Music features

In the present study, we have assessed the coupling between the EEG data and various properties of the musical stimuli. Of course, this required the extraction of such properties from the stimulus data in the first place. First, we defined a set of descriptors summarizing *low-level acoustic properties* of the music stimuli (**A**). Since the specific set of stimuli was monophonic, broadband envelope and fundamental frequency ($f_0$) of each note fully characterize the sound acoustics. However, only the envelope descriptor was used in the present study as the frequency information did not explain additional EEG variance. The broadband amplitude envelope was extracted from the acoustic waveform using the Hilbert transform (*Figure 1A*). In addition, **A** included the half-way rectified first-derivative of the envelope, which was shown to contribute to the stimulus-EEG mapping when using linear system identification methods (*Daube et al., 2019*).

In order to investigate the cortical processing of melodic expectations, we estimated *melodic surprise* and *entropy* for each individual note of a given musical piece by using IDyOM. Given a note $e_i$, a note sequence $e_{1..n}$ that immediately precedes that note, and an alphabet $E$ describing the possible pitch or note-onset values for the note, *melodic surprise* $S(e_i|e_{1..i-1})$ refers to the inverse probability of occurrence of a particular note at a given position in the melody. In other words, this surprise indicates the degree to which a note appearing in a given context in a melody is unexpected, or information content (*Pearce et al., 2010b*; *MacKay and Mac, 2003*):

$$S(e_i|e_{1..i-1}) = log_2 \frac{1}{p(e_i|e_{1..i-1})} .$$

The second feature that was extrapolated from IDyOM is the entropy in a given melodic context. This measure was defined as the Shannon entropy (*Shannon, 1948*) computed by averaging the surprise over all possible continuations of the note sequence, as described by E:

$$H(e_{1..i-1}) = \sum_{e \in E} p(e|e_{1..i-1}) S(e|e_{1..i-1}) .$$

Inverse probability and entropy are complementary in that the first indicates the level of expectedness of a note, while the second clarifies whether an unexpected note occurred in a context that

was more or less uncertain, thus corresponding to a weaker or stronger note sequence violation respectively.

IDyOM simulates implicit melodic learning by estimating the probability distribution of each upcoming note. This model can operate on multiple viewpoints, meaning that it can capture the distributions of various properties of music. Here, we focused on two such properties that are considered the most relevant to describe a melody: the *pitch* and the *onset time* of a note. IDyOM generates predictions of upcoming musical events based on what is learned, allowing the estimation of surprise and entropy values for the properties of interest. This provided us with four features describing the prediction of an upcoming note: surprise of pitch ($S_p$), entropy of pitch ($H_p$), surprise of onset time ($S_o$), and entropy of onset time ($H_o$). Each of these features was encoded into time-series by using their values to modulate the amplitude of a note-onset vector that is vectors of zeros marking with value one all note onsets, with length matching that of the corresponding musical piece and with the same sampling frequency as the EEG (or ECoG) data. The matrix composed of the four resulting vectors is referred to as *melodic expectations* feature-set (**M**).

In order to assess and quantify the contribution of melodic expectations to the music-EEG mapping, the main analyses were conducted on **A** and the concatenation of **A** and **M** (**AM**). The rationale is that the inclusion of M will improve the fitting score if the EEG responses to music are modulated by melodic expectations that is if **M** describes dynamics of the EEG signal that are not redundant with A (*Figure 1C*).

## Control analysis

The concatenation of acoustic and melodic expectation features **AM** constitutes a richer representation of a musical piece than **A** or **M** alone, and we hypothesized that it would be a better descriptor of the neural responses to music. However, it is also true that **AM** has more dimensions than **A**, which could be a confounding factor when comparing their coupling with the neural signal. In order to factor out dimensionality from this comparison, we have built stimulus descriptors with the same dimensionality as **AM** that carry the same acoustic information but less meaningful melodic expectation values, the hypothesis being that such descriptors would be less coupled with the neural signal. Such vectors were obtained by degrading the STM model by imposing memory restrictions on the local musical piece, while leaving untouched the LTM model, which represents the participants' prior knowledge on Western music. The memory restrictions on the STM model were introduced by subdividing each musical piece in chunks of exponentially longer lengths (1, 2, 4, 8, 16, and 32 musical bars) and then calculating the melodic expectations in each chunk separately. Similar results were obtained by reducing the model order *k*. However, the model order restricts the memory in terms of number of notes, while the first approach works in the musical bar dimension, which we considered more relevant and comparable across musical pieces.

The same analysis was conducted by using a stimulus descriptor consisting of the concatenation of **A** with the **M** descriptor after randomly shuffling the surprise and entropy values in time (but by preserving the note onset times; **AM**$_{shu}$), providing us with a feature-set with the same dimensionality and surprise- and entropy-values distributions of **AM** that contains **A** but not **M** information and that was outperformed by **AM** ($r_{AM} > r_A$: permutation test, $p<10^{-6}$, $d = 1.31$).

## Computational model and data analysis

A system identification technique was used to compute the channel-specific music-EEG mapping. This method, here referred to as the temporal response function (TRF; *Lalor et al., 2009*; *Ding et al., 2014*), uses a regularized linear regression (*Crosse et al., 2016*) to estimate a filter that optimally describes how the brain transforms a set of stimulus features into the corresponding neural response (forward model; *Figure 1B*). Leave-one-out cross-validation (across trials) was used to assess how well the TRF models could predict unseen data while controlling for overfitting. The quality of a prediction was quantified by calculating Pearson's correlation between the preprocessed recorded signals and the corresponding predictions at each scalp electrode.

The interaction between stimulus and recorded brain responses is not instantaneous, in fact a sound stimulus at time $t_0$ affects the brain signals for a certain time-window $[t_1, t_1+t_{win}]$, with $t_1 \geq 0$ and $t_{win} > 0$. The TRF takes this into account by including multiple time-lags between stimulus and neural signal, providing us with model weights that can be interpreted in both space (scalp

topographies) and time (music-EEG latencies). First, a time-lag window of −150–750 ms was used to fit the TRF models. The temporal dynamics of the music responses were inferred from the TRF model weights, as shown in *Figures 2E*, *3B* and *4B*. We then performed the EEG prediction analysis by restricting the TRF model fit to the window [0350] ms, thus reducing the dimensionality of the data and the risk for overfitting. This time-lag window was identified by means of a backward elimination procedure that quantified the relevance of the various stimulus-EEG latencies to the TRF mapping (*Figure 2—figure supplement 1*).

Backward elimination is a method to perform feature selection on multivariate data (*Guyon and Elisseeff, 2003*) (only the first iteration of this approach was run for computational reasons). In our context, the relevance of a feature (where feature includes both stimulus properties and time-lags) is quantified as the loss in EEG prediction correlation due to its exclusion from the TRF model. Specifically, TRF were fit for the time-lag window −150 and 750 ms after excluding a 50 ms window of time-lags $[t_i, t_i+50]$ ms. Then, the loss was calculated as $r_{LOSS} = r_{[-150,750]} − r_{[-150,750] \setminus [t_i, t_i+50]}$. Ultimately, this allowed us also to isolate the temporal dynamics of the effect of melodic surprise AM-A. Note that this procedure is similar to a single-lag analysis (*O'Sullivan et al., 2015*; *Das et al., 2016*), with the difference that latencies capturing information that is redundant with other lags will not produce a large $r_{LOSS}$, which is a useful property when the goal is to minimise the time-latency window. The results of this analysis are presented in *Figure 2—figure supplement 1*, where only significant $r_{LOSS}$ values are reported ($p < 0.05$, Bonferroni corrected permutation test with $N = 10000$).

### ERP analysis

To obtain time-locked neural responses to each note, the neural data were segmented and aligned according to note onset. Notes were grouped into *high* and *low* surprise by selecting the ones with the highest and lowest 20% $S_p$ values respectively. Among these epochs, we selected neural segments corresponding to notes with equal acoustic envelopes that is the 25% of notes with peak envelope amplitude closest to the median value (*Figure 5A*). ERPs and average acoustic envelopes were calculated by averaging the time-aligned neural data over each surprise group. Significant differences in the ERP traces between the two groups were calculated by means of an FDR-corrected permutation test. *Figure 5* shows the EEG result for channel Cz and the ECoG result for two selected electrodes. ERP power was calculated in the responsive latency window from 0 to 200 ms. ERP magnitude, which is often reported in μV, is indicated in arbitrary units (a.u.) here to avoid misleading the readers into comparing the absolute values of data from different recording modalities (EEG and ECoG).

### Statistical analysis

Statistical analyses were performed using two-tailed permutation tests for pair-wise comparisons. Correction for multiple comparisons was applied where necessary via the false discovery rate (FDR) approach. One-way ANOVA was used to assess when testing the significance of an effect over multiple (>2) groups (e.g., memory size in *Figure 2C*). The values reported use the convention $F(df, df_{error})$. Greenhouse-Geisser corrections was applied when the assumption of sphericity was not met (as indicated by a significant Mauchly's test). Cohen's *d* was used as a measure of effect size.

Topographical dissimilarity scores were calculated according to *Murray et al. (2008)*:

$$DISS = \sqrt{2*(1-r)}$$

where *r* is the correlation between two topographical distributions of interest. Significance was assessed by means of a one-sided *p*-values based on a randomization test with 100 permutations.

## Acknowledgements

Marcus Pearce and Jens Hjortkjær for useful discussion. Gaelle Rouvier for her help with the data collection. Part of the data analysis and discussions were conducted at the Telluride Cognitive Neuromorphic Engineering Workshop.

## Additional information

### Funding

| Funder | Grant reference number | Author |
|---|---|---|
| H2020 European Research Council | 787836 | Shihab Shamma |
| H2020 LEIT Information and Communication Technologies | 644732 | Alain de Cheveigné |
| National Institutes of Health | NIMH MH114166-01 | Ashesh D Mehta Nima Mesgarani |

The funders had no role in study design, data collection and interpretation, or the decision to submit the work for publication.

### Author contributions

Giovanni M Di Liberto, Conceptualization, Resources, Data curation, Software, Formal analysis, Validation, Investigation, Visualization, Methodology, Writing - original draft, Project administration, Writing - review and editing; Claire Pelofi, Conceptualization, Resources, Data curation, Formal analysis, Investigation, Writing - review and editing; Roberta Bianco, Resources, Data curation, Formal analysis, Methodology, Writing - review and editing; Prachi Patel, Resources, Data curation, Software, Formal analysis, Visualization, Methodology, Writing - review and editing; Ashesh D Mehta, Data curation, Funding acquisition, Methodology, Writing - review and editing; Jose L Herrero, Data curation, Writing - review and editing; Alain de Cheveigné, Conceptualization, Resources, Supervision, Funding acquisition, Methodology, Writing - review and editing; Shihab Shamma, Conceptualization, Resources, Supervision, Funding acquisition, Investigation, Writing - original draft, Project administration, Writing - review and editing; Nima Mesgarani, Conceptualization, Resources, Supervision, Funding acquisition, Investigation, Methodology, Project administration, Writing - review and editing

### Author ORCIDs

Giovanni M Di Liberto https://orcid.org/0000-0002-7361-0980
Ashesh D Mehta http://orcid.org/0000-0001-7293-1101
Nima Mesgarani https://orcid.org/0000-0002-2987-759X

### Ethics

Human subjects: Experimental procedures were approved by the CERES committee of Paris Descartes University (CERES 2013-11) and by the Feinstein Institute for Medical Research (07-125). All participants provided written informed consent before the experiment.

### Decision letter and Author response

Decision letter https://doi.org/10.7554/eLife.51784.sa1
Author response https://doi.org/10.7554/eLife.51784.sa2

## Additional files

### Supplementary files

• Supplementary file 1. Tables indicating the coordinates (MNI) of the intracranial electrodes for each patient.

• Supplementary file 2. Matlab interactive 3D plots showing the ECoG electrode placement for the first ECoG patient. Dots indicate ECoG channels. Red dots indicate channels that were responsive to the music input.

• Supplementary file 3. Matlab interactive 3D plots showing the ECoG electrode placement for the second ECoG patient. Dots indicate ECoG channels. Red dots indicate channels that were responsive to the music input.

• Supplementary file 4. Matlab interactive 3D plots showing the ECoG electrode placement for the third ECoG patient. Dots indicate ECoG channels. Red dots indicate channels that were responsive to the music input.

• Transparent reporting form

## Data availability

All EEG data and stimuli have been deposited on the Dryad repository. The TRF analysis was carried out using the freely available multivariate temporal response function (mTRF) toolbox, which can be downloaded from https://sourceforge.net/projects/aespa/.

The following dataset was generated:

| Author(s) | Year | Dataset title | Dataset URL | Database and Identifier |
|---|---|---|---|---|
| Giovanni M. Di Liberto, Claire Pelofi, Roberta Bianco, Prachi Patel, Ashesh D Mehta, Jose L Herrero, Alain de Cheveigné, Shihab Shamma, Nima Mesgarani | 2020 | Cortical encoding of melodic expectations in human temporal cortex | https://doi.org/10.5061/dryad.g1jwstqmh | Dryad Digital Repository, 10.5061/dryad.g1jwstqmh |

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
