## [Decision Letter]

**Acceptance summary:**

Predicting upcoming events is central to sensory experience, and expectation plays a fundamental role in processing music. Here the authors investigated neural responses to predictability in the context of natural music listening by presenting monophonic music to one group of listeners recorded with scalp EEG, and a second group with implanted electrodes. A model of melodic prediction based on Markov chains was used to estimate musical predictability. Auditory regions showed increased responses to expectations. Demonstrating cortical responses to expectation during is a useful extension to prior work, particularly in the context of natural listening. The work is of interest for its elegant investigation of expectation in music, but also has broader implications for prediction and expectation in sensory processing.

**Decision letter after peer review:**

Thank you for submitting your article "Cortical encoding of melodic expectations in human temporal cortex" for consideration by *eLife*. Your article has been reviewed by two peer reviewers, and the evaluation has been overseen by a Reviewing Editor and Barbara Shinn-Cunningham as the Senior Editor. The reviewers have opted to remain anonymous.

The reviewers have discussed the reviews with one another and the Reviewing Editor has drafted this decision to help you prepare a revised submission.

Summary

The authors investigated neural responses to predictability in the context of musical rhythm by presenting monophonic music to one group of listeners recorded with scalp EEG, and a second group with implanted electrodes (ECoG). A model of melodic prediction based on Markov chains was used to estimate musical predictability. Auditory regions (auditory cortex and planum temporale) showed increased responses to expectations. Demonstrating cortical responses to expectation during is a useful extension to prior work, particularly in the context of natural listening.

Essential revisions

1) The manuscript does not make adequate contact with existing literatures on music processing and the theoretical advances achieved by the current study. The Introduction explains the methods but does not situate the current results within a broader theoretical framework (for example, a few prior music studies are cited but not explained). Although the methods are generally strong, the theoretical framing must be strengthened substantially in both the Introduction and Discussion.

2) We do have substantial knowledge on which regions process musical structure – and those go beyond temporal areas (e.g., including IFG). The way the study is constructed and presented neglects possible contributions of these regions. ECoG analyses were limited to electrodes that showed auditory responses which biases results to the temporal lobe (as said at the end of the third paragraph of the Discussion section). Why this limitation? And/or why not fitting data to M only?

3) The paper is substantially lacking in the clarity of anatomical detail, particularly regarding iEEG data. Figure 3 and Figure 4 would benefit from anatomy panels depicting the location of recording sites in the subjects or some other way to visualize where the recordings were made from. It is important to have information about overall electrode coverage to estimate which parts of the perisylvian network were researchable.

4) The Discussion proposes "fundamental differences between music and speech perception" based on discrepancies between the present finding on musical melody (syntax) and previous findings on phonotactic and semantic processing in language. However, there are no theoretical grounds to compare these findings. At least linguists draw more or less clear borders between phonology, syntax and semantics and different neural networks are being discussed for these processes (plausible reason for different scalp topographies for example, Discussion paragraph five). A comparison of the present music data to data on syntax in language may seem more suitable (preferably as within-subject comparison).

---

## [Author Response]

Essential revisions1) The manuscript does not make adequate contact with existing literatures on music processing and the theoretical advances achieved by the current study. The Introduction explains the methods but does not situate the current results within a broader theoretical framework (for example, a few prior music studies are cited but not explained). Although the methods are generally strong, the theoretical framing must be strengthened substantially in both the Introduction and Discussion.

We agree with the reviewer that the previous version of the manuscript did not sufficiently link our study to the rich literature in music research. A significant part of the Introduction and Discussion sections have been rewritten to more clearly relate our work with that previous literature. The main changes include:

– An introduction on the theoretical views and computational models attempting to explain how our brains learn musical structures (Introduction paragraph one);

– A more comprehensive introduction of previous neurophysiological work, especially with a focus on ERP results (Introduction paragraph three);

– A more in-depth explanation of the limitations of previous studies that our experiment overcomes (Introduction paragraph four). We also detail the limitations of our approach and discuss how future work could further advance our understanding of the neural underpinnings of music perception (Discussion paragraph three).

– A more comprehensive and careful discussion on the similarities and differences between our findings on melodic expectations encoding and previous work on language processing (as the reviewer pointed out in the fourth essential point) (Discussion paragraph five).

The link between our study and previous research has also been generally reinforced throughout the paper with additional references and considerations.

2) We do have substantial knowledge on which regions process musical structure – and those go beyond temporal areas (e.g., including IFG). The way the study is constructed and presented neglects possible contributions of these regions. ECoG analyses were limited to electrodes that showed auditory responses which biases results to the temporal lobe (as said at the end of the third paragraph of the Discussion section). Why this limitation? And/or why not fitting data to M only?

We thank the reviewers for raising this issue. This question made us realise that our explanation of the adopted methodology was unclear, thus leading us to apply major changes to the manuscript.

ECoG analyses were conducted on electrodes which showed stronger responses to note-onsets than to silence. While this approach certainly selected auditory responsive electrodes in temporal cortex, any other response that was time-locked to note-onset would have emerged as well, thus including melodic expectation responses that were time-locked to note onset. Note that this is the same assumption of our TRF analysis which, in fact, requires time-locking between the stimulus (note sequences) and the neural responses. In other words, our electrode selection approach identified electrodes capturing time-locked responses, which are also the same ones where the TRF analysis can be effective. As the reviewer suggested, another option could be to perform the channel selection directly on the TRF results. However, this comes at the risk of introducing a bias towards the particular set of features used for the model fit (A, M, AM, AM-A).

Regarding the possible contribution of areas beyond temporal cortex, our coverage of relevant areas such as IFG was not sufficient to claim neither a positive nor a null result on this (we found one responsive electrode in IFG). For this reason, the revised manuscript focuses on our findings in the temporal cortex but also reports the results and anatomical locations of other relevant electrodes (see Figures 3C, 4C, Figure 3—figure supplement 1 and Figure 4—figure supplement 1, Supplementary file 1). In addition, we also report on the full coverage for each ECoG patient with interactive 3D brain plots (Supplementary files 2-4).

We have clarified these points in the text, both in the Materials and methods, Results, and Discussion sections.

3) The paper is substantially lacking in the clarity of anatomical detail, particularly regarding iEEG data. Figure 3 and Figure 4 would benefit from anatomy panels depicting the location of recording sites in the subjects or some other way to visualize where the recordings were made from. It is important to have information about overall electrode coverage to estimate which parts of the perisylvian network were researchable.

Anatomy panels indicating the electrodes selected for the analysis have been added to Figures 3, 4, Figure 3—figure supplement 1 and Figure 4—figure supplement 1 (note that the exact location of those electrodes is specified in the supplementary files). Furthermore, we have added supplementary interactive 3D plots with the full coverage of the recordings that 1) give a more precise graphical indication on where the selected electrodes were positioned and 2) provide information also on the electrodes that were not sensitive to our analyses.

4) The Discussion proposes "fundamental differences between music and speech perception" based on discrepancies between the present finding on musical melody (syntax) and previous findings on phonotactic and semantic processing in language. However, there are no theoretical grounds to compare these findings. At least linguists draw more or less clear borders between phonology, syntax and semantics and different neural networks are being discussed for these processes (plausible reason for different scalp topographies for example, Discussion paragraph five). A comparison of the present music data to data on syntax in language may seem more suitable (preferably as within-subject comparison).

Syntax is certainly a critical part of language processing that has often been the focus of studies that looked for shared mechanisms between speech and music processing. In this context, it is also important to investigate various other levels of speech processing (e.g. prosody) (Heffner and Slevs, 2015). The revised version of the paper expands this discussion by pointing to similarities between our results and previous work on language perception at various processing levels. This part of the manuscript has been largely rewritten, especially by including a discussion on previous language research on syntax processing. We also clarified that similarities between our results and previous research should be taken only as speculations for now, and that further work should be conducted to explicitly compare the cortical underpinnings of music and language perception.